# *Vampyrum spectrum* (Phyllostomidae) movement and prey revealed by radio-telemetry and DNA metabarcoding

José Gabriel Martínez-Fonseca[1,2]*, Rebecca Mau[2], Faith M. Walker[1,2], Arnulfo Medina-Fitoria[3], Kei Yasuda[4], Carol L. Chambers[1]

1 Bat Ecology & Genetics Lab, School of Forestry, Northern Arizona University, Flagstaff, Arizona, United States of America, 2 Pathogen and Microbiome Institute, Northern Arizona University, Flagstaff, Arizona, United States of America, 3 Asociación mastozoológica de Nicaragua (AMAN), Ticuantepe, Nicaragua, 4 Independent field research assistant, Eugene, Oregon, United States of America

* jm3934@nau.edu

**Data Availability Statement:** All raw sequences were submitted to NCBI SRA (submission ID

## Abstract

The spectral bat (*Vampyrum spectrum*), the largest bat species in the Americas, is considered Near Threatened by the International Union for Conservation of Nature and is listed as a species of special concern or endangered in several countries throughout its range. Although the species is known as carnivorous, data on basic ecology, including habitat selection and primary diet items, are limited owing to its relative rarity and difficulty in capturing the species. Leveraging advances in DNA metabarcoding and using radio-telemetry, we present novel information on the diet and movement of *V. spectrum* based on locations of a radio-collared individual and fecal samples collected from its communal roost (three individuals) in the Lowland Dry Forest of southern Nicaragua. Using a non-invasive approach, we explored the diet of the species with genetic markers designed to capture a range of arthropods and vertebrate targets from fecal samples. We identified 27 species of vertebrate prey which included birds, rodents, and other bat species. Our evidence suggested that *V. spectrum* can forage on a variety of species, from those associated with mature forests to forest edge-dwellers. Characteristics of the roost and our telemetry data underscore the importance of large trees for roosting in mature forest patches for the species. These data can inform conservation efforts for preserving both the habitat and the prey items in remnants of mature forest required by *Vampyrum spectrum* to survive in landscape mosaics.

## Introduction

With a mass of up to 235 g, the spectral bat (*Vampyrum spectrum*) is the largest bat species on the American continents [1, 2]. This species belongs to the subfamily Phyllostominae, the leaf-nosed bats (Phyllostomidae), which inhabit forests with relatively low anthropogenic disturbance [3–7]. The International Union for Conservation of Nature (IUCN) has designated the species as Near Threatened with a decreasing population trend throughout its range [8]. The species is considered at risk of extinction in several countries, including Bolivia, Mexico, and

SUB10749581, BioProject ID PRJNA785628, accession numbers SAMN23572418-51).

**Funding:** The authors received no specific funding for this work.

**Competing interests:** The authors have declared that no competing interest exist.

Nicaragua [9, 10]. In Nicaragua, *V. spectrum* was first recorded in 1910 and currently only a handful of capture sites are known, mainly in the Pacific and the southeastern regions of the country [11, 12]. Despite its widespread distribution on the continent, the basic ecology of the species is largely unknown [13].

Based on roost descriptions from Costa Rica, Mexico, Trinidad and Tobago, the species selects large hollow tree trunks in primary forests [2, 7]. Foraging and activity outside the roost seems to be irregular and might correspond to resource availability [2]. Mammals and birds constitute much of the diet of the species [13, 14]. Several studies recorded predation of rodents and other bat species based on prey remains (e.g., bones, wing and tail parts) at the bottom of roosts and trials with captive animals [15–17]. Feathers found at a communal roost entrance suggest the predation and transport of at least 18 bird species ranging from the 150 g White-tipped Dove (*Leptotila verreauxy*) to the 20 g Banded Wren (*Thryothorus pleurosticus*) [2]. In some cases, examination of stomach contents revealed the bones of bats, passerine birds, and arboreal rodents [13, 18, 19]. Visual examination of stomach contents and remains at roosts have obvious constraints on the ability to recognize diet items and often implies that collection of live specimens is necessary (e.g., [13]). With genetic approaches, one can identify items that have been partially or wholly digested (i.e., non-identifiable from stomach contents), exist in scarce quantities, or were eaten in a previous meal [20]. In recent years, advances in genetic techniques have facilitated the study of diet using fecal samples from a diversity of animal species including bats (e.g., [21, 22]). In Panama, metabarcoding techniques confirmed that *Trachops cirrhosus* fed not only on a diversity of frogs but also small mammals and reptiles [23]. However, in the Neotropics, DNA techniques for bat diet determination is a novel area for research.

The goals of this study were to identify the diet of *V. spectrum* and to describe roost and movement in the Lowland Dry Forest (LDF) of the Pacific coast of Nicaragua. We used habitat information for prey species identified in the diet of *V. spectrum* to infer forest conditions used by this predator while foraging and commuting. Additionally, we discuss management and conservation implications of our findings for this species.

## Methods

### Study site

Our study site encompassed the > 2000 ha Escameca Grande Private Reserve and surrounding forested lands in Rivas Department, Pacific of Nicaragua (11°12'33" N, 85°44'2" W). The Rivas Isthmus (~70 km long, 40–18 km wide) separates the Pacific Ocean and the south coast of Lake Nicaragua and includes a mosaic of pastures, seasonal and permanent crops, and remnants of mature and secondary growth of deciduous LDF [24–26]. The area experiences strong seasonality with well-marked rainy (mid-May to the end of November) and dry (remainder of year) seasons. Annual precipitation varies from 1400 to 2000 mm with areas near the Lake Nicaragua receiving higher precipitation. Terrain in the isthmus is rougher near the Pacific coast (elev. ≤400 m) with a low-elevation plain near the lake shore (elev. ~40 m) [27].

### Live capture and telemetry

We used single and triple high mist nets (Avinet Research Supplies, USA) to capture bats along flyways and over water [28]. Nets were open at ~1750 h and closed at ~0030 h. Bats were handled under Nicaraguan permit of the Ministerio de Ambiente y Recursos Naturales (MARENA) No.015-122011, with the approval of the Northern Arizona University Institutional Animal Care and Use Committee (15–006) and under guidelines of the American Society of Mammologists [29]. To identify species, we used a field key and descriptions by Reid [1], Medina-Fitoria

[30], and nomenclature according to Martínez-Fonseca et al. [12]. Standard morphometric measurements (mass, forearm length, sex, reproductive status) were collected from each individual.

We used two methods for attaching the radio transmitters to maximize the chances of at least one transmitter staying on the bat long enough to track the bat to roost sites. In previous attempts, we attached transmitters to large bats (e.g., *Phylloderma stenops*, *Phyllostomus hastatus*) that fell off or bats removed in ≤ 2 nights. We applied two radio transmitter models (BD2 [0.9 g], BD2C [1.1 g], Holohil Systems Ltd., Canada) to an individual to locate day roost. One transmitter was a custom-made collar designed to release after ~3 weeks and the second was designed to be attached between the scapula using a non-toxic medical glue (Torbot Liquid Bonding Cement, Torbot Group Inc., USA). We radio tracked using a close approach method [31] with a receiver (R1000, Communication Specialist Inc., USA) and two-element Yagi antenna (RA-23K, Telonics Inc., USA). We recorded all locations in geographic coordinate system WGS84 with a precision of <3 m using a handheld Global Positioning System device (Garmin 64map, Garmin Ltd., Switzerland). To confirm locations of the roost cavities, we climbed trees identified as potential roosts (based on telemetry signal) and visually inspected cavities. To record exit and entry time of the radio tagged animal from the roost, a team member monitored the signal and position of the animal with a receiver as frequently as possible at night (from 1730 h to 0630 h). To avoid disturbing the animals and affecting their behavior, observations were made at >30 m from the base of the roost tree.

## Fecal DNA collection and analysis

For diet analysis, we collected 10 subsamples of fecal material upon initial discovery of the roost (Sample A). Additionally, six subsamples of fresh pellets (Sample B; defecated after discovery of the roost) were collected by placing a clear plastic sheet at the bottom of the roost and retrieving the freshly deposited guano the following day. Samples were taken from directly under the bats and constituted several guano pellets of about 2.5 ml in total volume. We only collected fresh samples one night to avoid disturbance to animals and reduce interference with their normal activity patterns.

Fecal samples were stored in 15 mL conical tubes containing 7 mL RNAlater solution (Thermo Fisher Scientific Inc., USA). Once transported to the USA, fecal samples were stored at -20°C in the Bat Ecology and Genetics lab at Northern Arizona University (https://www.nau.edu/sff). We extracted DNA from homogenized samples using a QIAamp Fast DNA Stool Mini Kit (Qiagen Inc., USA) following manufacturer instructions. We performed initial PCRs using four primers that targeted identification of a range of vertebrates and arthropods: ANML [32], 12S [33], 18S [34] and SFF [35, 36]. Each primer set was run on replicates of the 16 samples on its own plate with non-template (NTC) and positive control (PTC) for a total of 72 wells across four plates. We followed the corresponding PCR protocols and cycling conditions for each primer and used an Applied Biosystems SimpliAmp thermocycler (Thermo Fisher Scientific Inc., USA).

We removed PCR primer by-products with a 1 × bead cleanup using AMPure XP beads (Beckman Coulter Inc., USA). Indexing barcodes [37] were added with the KAPA HiFi (KAPA Biosystems Inc., USA) master mix. We quantified DNA concentrations using KAPA Library Quantification Kit (KAPA Biosystems, USA) and pooled equimolar samples for sequencing on a MiSeqGA2x with a V3 600 cycle kit (Illumina Inc., USA). All samples from all maker sets were combined in the same run.

We processed sequencing data using the QIIME2 pipeline [38]. Primers were trimmed off sequences using cutadapt [39] and paired-end reads were trimmed, denoised, joined, dereplicated, and chimera filtered using the DADA2 [40] command in QIIME2 (qiime dada2

denoise-paired). Amplicon Sequence Variants (ASVs) from the primers targeting the 12S rRNA gene were trimmed to 150bp during the DADA2 step, 18S rRNA amplicons were trimmed to 100bp, amplicons from the ANML primers targeting the COI gene were trimmed to 110bp, and amplicons from the SFF primers were trimmed to 130bp and 115bp for the forward and reverse reads, respectively. To increase confidence in diet item sequences, we required the ASV to be present in at least 3 samples and filtered out any ASVs that did not meet this requirement. We assigned taxonomy using the NCBI's Basic Local Alignment Search Tool (BLAST) and database (https://blast.ncbi.nlm.nih.gov/). Since many species of vertebrates with recent taxonomical splits in Nicaragua do not have reference sequences in GenBank, we assigned IDs to sister species known to occur in our study area. For ranges in mass and ecological data of vertebrate species in *V. spectrum* fecal samples, we used Reid [1], Chavarría-Duriaux et al. [41], and Billerman et al. [42].

## Results

### Roost and activity patterns

On 3 March 2017, at 0000 h, we caught a post-lactating female *V. spectrum* (mass = 160 g, FA = 116 mm, E = 40 mm, RF = 28 mm) at ground level in the bottom shelf of a single 12 m x 2.7 m mist net. The net was located across a dry riverbed of a small tributary of the Escameca Grande River, San Juan del Sur (11°11'31" N, 85°46'6" W, 47 m). Other species captured at the same site included *Artibeus jamaicensis, A. lituratus, A. phaeotis, Micronycteris microtis, Carollia perspicillata, C. subrufa, Glossophaga commissarisi, Phyllostomus discolor, Sturnira parvidens, Saccopteryx bilineata,* and *Tonatia bakeri.*

The animal was released at the capture site following attachment of two transmitters once we ensured appropriate collar fit. Both radio transmitters combined accounted for <5% of the mass of the bat before release. We radio tracked the animal the following day to the roost area located 720 m east of the capture location. The density of standing trees, their foliage, and the fact that the bat was enclosed in the interior of a live tree affected signal radiation and, therefore, reception to ~250 m. Because of personnel and climbing equipment constraints, identifying the specific roost tree was delayed to 14 March, when a member of our team was able to climb and identify the roost entrance 12.5 m above ground in a guayabón tree (*Terminalia oblonga*). The roost tree was 66.5 cm diameter at breast height and 27 m tall. The roost tree contained other cavities and was located 15 m from the riverbed of the Escameca Grande.

Once the roost entrance in the tree was identified, we confirmed presence of *V. spectrum* inside the roost, and collect 10 subsamples "A "and came back the next day to collect six more fecal subsamples "B". The roost cavity had a "boot" shape, with an entrance of about 20 cm in diameter that expanded to 30 cm inside. The ceiling of the roost was about 1.2 m above the entrance (Fig 1A). We observed three individual *V. spectrum* of approximately the same size inside (family unit). The transmitter collar was not visible because of the position (close proximity) of the three individuals (Fig 1B). The bats were observed at the upper area of the cavity and did not move during our observations. White-tipped dove (*Leptotila verreauxi*) feathers were recognizable at the bottom of the roost atop a guano pile, and we assumed them to be remnants of a prey item (Fig 1C). The guano pile itself was 10 cm deep, relatively flat due to decomposition of the older depositions, and filled the base of the roost except for the entrance. Recent depositions were of a lighter color than the older, more decomposed ones. Additionally, spine-like hairs of Salvin´s pocket mice (*Liomys salvini*) were recognizable in the fresh and old guano. We found no indication that other species (e.g., bat, arboreal mammals, or birds) used the cavity concurrently with the *V. spectrum.*

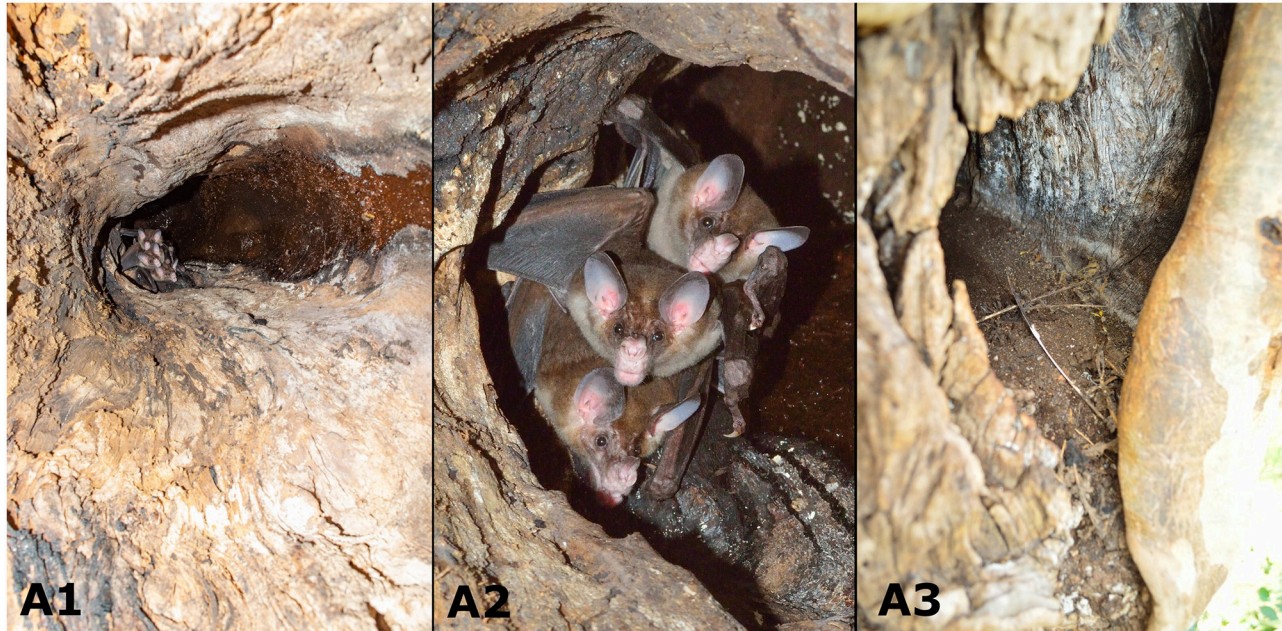

**Fig 1. Roosting *Vampyrum spectrum*.** Wide angle view of the roost of a radio-tagged post-lactating female (A), close-up of the three individuals roosting presumed to be a family unit (B), and entrance to the tree cavity (C), Rivas, Nicaragua, March 2017.

Activity monitoring of the radio-tagged individual was conducted opportunistically (based on transportation and personnel availability) during 11 non-consecutive nights between 14 March to 15 April 2017. The bat left the roost once per night during each of the observation nights. The bat spent 37 to 280 minutes out of the roost each night (mean and SE: 127 ± 20.8 minutes; Table 1). The limited range of the transmitter signal in the forest environment and limited road access in the area made impossible to track the bat during foraging at night. On 16 March, the transmitter that was glued to the back of the *V. spectrum* was found 160 m away from the riverbed of the Escameca Grande River, 1.6 km southwest from the original capture

**Table 1. Activity log (exit and return times) of a post-lactating female *Vampyrum spectrum* in the Lowland Dry Forest of the southeastern Pacific coast of Nicaragua.**

| Date | Exited (h) | Returned (h) | Duration (min) |
|---|---|---|---|
| 14-Mar-17 | 0021 | 0230 | 141 |
| 22-Mar-17 | 0215 | 0432 | 137 |
| 23-Mar-17 | 2347 | 0213 | 146 |
| 01-Apr-17 | 2234 | 0102 | 148 |
| 05-Apr-17 | 2034 | 2323 | 50 |
| 06-Apr-17 | 2157 | 2234 | 37 |
| 07-Apr-17 | 2043 | 2250 | 113 |
| 08-Apr-17 | 2105 | No data collected | NA |
| 13-Apr-17 | 1942 | 2152 | 110 |
| 14-Apr-17 | 2032 | 2222 | 110 |
| 15-Apr-17 | 1923 | 0003 | 280 |

Mean and SE = 127.2 ± 20.89 min.

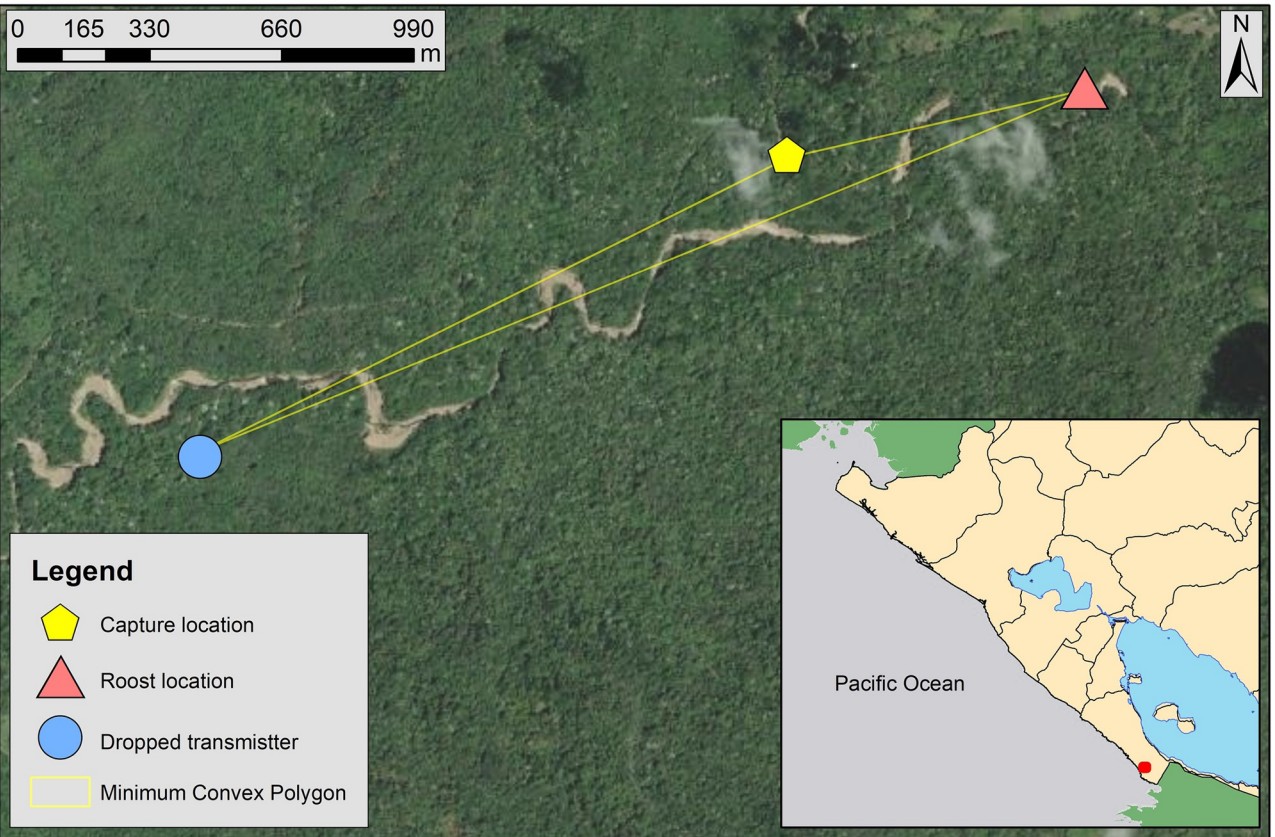

**Fig 2. Telemetry locations post-lactating female spectral bat (*Vampyrym spectrum*) in Escameca Grande Private Reserve, Rivas Department, Nicaragua.** Capture, roost, and dropped transmitter. The area of the minimum convex polygon is 16.3 ha. The red dot in the inset map shows the general study location in the southeastern Pacific coast of Nicaragua. Basemap layer from ArcGIS Online maps under a CC BY license, with permission from Esri, original Copyright 2018 Esri (Basemaps supported by Esri, DigitalGlobe, GeoEye, Earthstar Geographics, CNES/Airbus Ds, USDA, AEX, Getmapping, Aerogrid, IGN, IGP, swisstopo, and the GIS User Community).

location and 2.7 km from the roost. The area enclosed by the roost, capture site, and dropped transmitter locality was 16.3 ha (Fig 2). The transmitter in the collar stayed attached to the animal until the signal was too unreliable to monitor and vanished (probably due to battery failure) on 16 April 2017.

## Diet

We detected 27 species of terrestrial vertebrates in fecal samples representing birds and mammals, including the host species (Table 2). Sample B contained fewer vertebrate species (n = 5) compared to the Sample A (n = 25). Birds constituted the largest group, with 12 native and one non-native bird species. Maximum mass of prey ranged from 32 g (*Tyrannus savanna*) to 170 g (*Megascops cooperi*). We also detected domestic chicken (*Gallus gallus*; which range from 35–65 g in chicks to 1–2.5 kg adults).

Bats represented the second largest group in species richness for *V. spectrum* diet. We detected 11 species of bats excluding the host DNA. Maximum mass for bat species ranged from 5 g (*Rhogeessa bickami*) to 59 g (*Eumops underwoodi*) and species encompassed nine genera. The third group of diet items contained three terrestrial rodents *Liomys salvini* (65 g), *Sigmodon hispidus* (130 g), and *Sphiggurus mexicanus* (~2600 g). *Liomys salvini* was the only

**Table 2. Vertebrate species identified in fecal samples collected from a *Vampyrum spectrum* roost using four genetic markers (ANML, SFF, 12S, 18S), Rivas department, Nicaragua 2017.**

| Taxa | Common name | Habitat | Mass (g) | Sample A | | | | Sample B | | | |
|---|---|---|---|---|---|---|---|---|---|---|---|
| | | | | ANML | SFF | 12S | 18S | ANML | SFF | 12S | 18S |
| **Birds** | | | | | | | | | | | |
| *Antrostomus carolinensis* | Chuck-will's-widow | G | 42–69 | | | 6 | | | | | |
| *Campylorhynchus* [*rufinucha*] *capistratus*[a] | Rufous-backed Wren | G | 30–35 | | 4 | | | | | | |
| *Coccyzus minor* | Mangrove Cuckoo | S | 64–102 | | 2 | 6 | | | | | |
| *Columbidae* | | | | | | 6 | | | | | |
| *Crotophaga sulcirostris*[a] | Groove-billed Ani | G | 70–90 | | | 6 | | | | | |
| *Gallus gallus* | Domestic chicken | | 35–2500 | | 2 | | | | | | |
| *Leptotila verreauxi*[a] | White-tipped Dove | G | 96–155 | | 5 | 10 | | | | | |
| *Megascops* [*cooperi*] *kennicotti* | Pacific Screech-owl | G | 147–170 | 1 | | | | | | | |
| *Melanerpes* [*hoffmanni*] *aurifrons* | Hoffmann's Woodpecker | G | 32–84 | | 1 | | | | | | |
| *Myiarchus crinitus* | Great Crested Flycatcher | G | 27–40 | | 3 | | | 2 | | | |
| *Notharchus* [*macrorhynchos*] *hyperrhynchus* | White-necked Puffbird | G | 81–106 | | 3 | | | | | | |
| *Passeriformes*[b,c] | | | | | | 10 | | | | | |
| *Piaya cayana* | Squirrel Cuckoo | G | 98–110 | | 1 | 5 | | | | | |
| *Tyrannus savanna* | Fork-tailed Flycatcher | G | 28–32 | | | 8 | | | | | |
| **Mammals** | | | | | | | 5 | | | | 2 |
| *Chiroptera*[d] | | | | | | | | | | | |
| *Artibeus*[c] | | | | 2 | | | | | | | |
| *Artibeus jamaicensis* | Jamaican fruit-eating bat | G | 29–51 | | 2 | 4 | | | | | |
| *Carollia perspicillata* | Seba's short-tailed bat | G | 15–25 | | | | | | 1 | | |
| *Desmodus rotundus* | Common vampire bat | G | 19–43 | | 2 | 7 | | | | | |
| *Diaemus youngi* | White-winged vampire bat | S | 32–40 | | 1 | 3 | | | | | |
| *Eumops underwoodi* | Underwood's bonneted bat | G | 58–59 | | 1 | | | | | | |
| *Micronycteris* [*microtis*] *megalotis* | Common big-eared bat | S | 4–9 | | 2 | 3 | | | | | |
| *Micronycteris minuta* | White-bellied big-eared bat | S | 4–8 | | | 3 | | | | | |
| *Micronycteris schmidtorum* | Schmidt´s big-eared bat | S | 5–8 | | 2 | | | | | | |
| *Phyllostomidae* | | | | | | 4 | | | | 5 | |
| *Phyllostomus discolor* | Pale spear-nosed bat | G | 26–51 | | | 7 | | | | | |
| *Pteronotus davyi fulvus* | Davy's naked-backed bat | G | 5–10 | | 1 | | | | 2 | | |
| *Rhogeessa* [*bickhami*] *tumida* | Bickham's little yellow bat | G | 3–5 | 1 | 2 | 5 | | | | | |
| *Vampyrum spectrum*[e] | Spectral bat | S | 135–235 | | | 8 | 10 | | 6 | 5 | |
| **Terrestrial mammals** | | | | | | | | | | | |
| *Heteromyidae*[c] | | | | | | 9 | | | | 4 | |
| *Liomys salvini* | Salvin's spiny pocket mouse | G | 30–65 | 5 | 9 | 10 | 5 | 5 | 6 | 5 | 3 |
| *Sigmodon hispidus* | Hispid cotton rat | G | 38–130 | | | 6 | | | | | |
| *Sphiggurus mexicanus*[e] | Central American porcupine | G | 1400–2600 | | | 10 | | | | | |

Samples were pooled from a roost that supported three individual bats; values indicate the number of samples in which the species was detected. Ten subsamples were collected when the roost was first identified (Sample A); six additional subsamples during a 1-night period after locating the roost (Sample B). Habitat association includes habitat generalist (G) and habitat specialist (S). Scientific names in brackets indicate the updated taxonomy for Nicaraguan populations. Range in mass (g), common names, and habitat associations are according to Reid [1], Chavarría-Duriaux et al. [41] and Billerman et al. [42].

[a] Previously described diet items by Vehrencamp et al. [2].

[b] Previously described diet items by Casebeer et al. [18].

[c] Previously described diet items by Bonato et al. [13].

[d] Previously described diet items by Peterson and Kirmse [19].

[e] Species that we do not consider to be part of the diet.

species consistently detected across all 16 subsamples and four genetic markers. Several invertebrate, bacterial, and fungal groups were detected by the ANML, SFF, and 12S primers.

Arthropod groups detected included Arachnida, Blattodea, Coleoptera, Diptera, Hymenoptera, and Lepidoptera. Bonato et al. [13] reported arthropods in the diet of *V*. *spectrum* but did not clarify how they discriminated items consumed by the prey, an issue documented by Sheppard et al. [43]. There is also a likelihood that many invertebrate items originated from organisms living in the guano, or the tree. Additionally, the incompleteness of sequence libraries for Central American arthropods limits the confidence of identification of potential diet items at this time. Here, we focused our reporting and discussion on the vertebrates we identified. As reference libraries become more complete a future re-analysis of the data and provide informative results, therefore, all raw sequences from all markers were submitted to NCBI SRA (submission ID SUB10749581, BioProject ID PRJNA785628, accession numbers SAMN23572418-51).

## Discussion

Considering the diversity of prey species and variation in foraging duration, we believe that *V*. *spectrum* forage opportunistically on vertebrates that included both habitat specialists and generalists. All species that we identified in the fecal samples of *V*. *spectrum* were known to occur in the study area [1, 12, 41]. Twenty-five of the vertebrate species detected in the fecal samples fall within the mass range of previously documented prey and we considered them diet items for *V*. *spectrum* [2, 7, 13, 17–19]. The SFF marker, which is optimized for identifying bat species from fecal samples [35, 36], followed by the 12S marker [33], provided the greatest diversity of identifiable vertebrate taxa with 10 species confirmed by both markers. The ANML primer [32] detected fewer vertebrate taxa; however, it was the only one that detected *Megascops cooperi*. This highlights the importance of using a multiple set of markers to maximize the detection of prey items.

Birds may represent an energetically cost-effective and abundant food source for *V*. *spectrum*. Species like *Campylorhynchus capistratus*, *Coccyzus minor*, *Crotophaga sulcirostris*, and *Zenaida asiatica* were already reported as part of the diet [2]. Excluding *M. cooperi* and *Antrostomus carolinensis*, all birds detected in the samples were diurnal species which might suggest that they were captured while at rest and required less energy to obtain. In addition, bird species detected are known to have diverse habitat associations. In Nicaragua, species like the mangrove cuckoo (*C. minor*) are mostly tied to the proximities of wetlands, riparian forests, and estuaries, while *Northarchus macrorhynchus*, *Piaya cayana*, and *Campylorhynchus rufinucha* are common on forest edges, secondary growth, and urbanized areas [41]. However, because the habitat descriptions of these birds corresponded to diurnal observations and not nocturnal roosts, they might not reflect the actual environment in which they were captured [44].

Unlike birds, all mammal species detected in the fecal samples of *V*. *spectrum* are predominantly nocturnal or are more active at dusk [1]. Bat species represented most of the mammal species detected in our samples and represented a range of masses, foraging guilds, and habitat associations. Species like *Micronycteris microtis*, *M. minuta*, *M. schmidtorum*, *Pteronotus davyi*, and *Rhogeessa bickami* are small insectivorous bats that either glean or catch their prey during flight [45–48]. *Micronycteris* species are often tied to mature, less-disturbed, forest conditions and are captured in small numbers in this area [26, 49]. *Artibeus jamaicensis* and *Carollia perspicillata* are frugivores, abundant along forest edges and within disturbed forests; they are also the two most abundant species in the area [26, 50–52]. The two vampire bats detected were exclusively hematophagous with *Desmodus rotundus* being the third most abundant species in the disturbed landscape matrix of our study area due the presence of cattle (*Bos* spp.;

[26]). It is not clear how *V. spectrum* hunts and captures other bats [14, 19]. It is possible that the species can prey upon bats while they are at rest in foliage or a roost, but aerial hawking of other bat species has been observed. In Panama, *V. spectrum* was observed, while in flight, using one of its wings to strike and manipulate a flying *Sturnira parvidens* into its mouth (F. J. Bonaccorso, pers. comm.).

*Liomys salvini* might constitute the most common terrestrial mammal in the diet of *V. spectrum* and was detected in all our samples. This species of pocket mouse is abundant in mature and secondary forest in the LDF of Nicaragua and are often observed in our study area, supporting the idea that *V. spectrum* hunts opportunistically on available vertebrates species ([1]; Medina-Fitoria pers. comm.). Similarly sized rodents are also reported in stomach contents of *V. spectrum* from Panama [7, 19].

The only two species detected that did not matched mass ranges for previously described prey were *S. mexicanus* and domestic chicken [2, 7, 18]. *Sphiggurus mexicanus* is a common semiarboreal mammal species in the LDF of the Rivas Department and is often observed sharing roosts with bat species including *Carollia perspicillata*, *Tonatia surophila*, and *Saccopteryx bilineata* (A. Medina-Fitoria and J. G. Martínez-Fonseca pers. obs.). The detection of porcupine DNA was possibly from an individual searching for a tree cavity as a potential rest site. The DNA from domestic chicken could indicate that *V. spectrum* preyed on young birds (a village where chickens were common was <600 m from the roost). Both vampire species that we detected fed on blood from domestic birds and cattle [53, 54]. Although we did not detect cattle in our samples, Walker et al. [35] genetically identified cattle from *D. rotundus* feces, so it is not unrealistic to conclude that domestic birds could be either prey of *V. spectrum* or reflect diet of vampire bats that we detected.

Other "false vampire" bat species like *Macroderma gigas*, *Megaderma lyra*, and *M. spasma* from Australia and Asia prey on amphibians and reptiles [55–57]. However, we did not detect these despite using 12S and 18S markers which other studies have successfully used for barcoding these taxa (e.g., [58–60]). Additionally, our study was conducted during the dry season which could negatively impact the availability of these prey items for *V. spectrum*.

The lower diversity of diet items in Sample B compared to Sample A likely reflected that the latter represented accumulation of fecal material over a longer period, and thus more prey species. Walker et al. [35] successfully sequenced DNA in six-month-old bat fecal samples in Belize, and found that dry conditions can help preserve DNA for longer periods of time. The dry season, during which our study was conducted, might have helped reduce degradation of DNA in the feces for weeks or months.

The spatial data obtained from our radio tagged *V. spectrum* indicated that it used tall riparian forest areas of the LDF of Nicaragua. The sites for the roost, capture, and dropped transmitter matched assumptions of the species' preference for medium to large patches of mature, tall canopy forests that are relatively undisturbed [2, 8, 18, 19]. This corresponded with the previous knowledge of the species in other forest types but the distance between two of our localities was 2.4 km, implying that the home range of the species is considerably greater than the 3.2 ha previously reported [2]. Both hunting and resting space (large, tall trees, and dense canopy) and safe roost sites (needed for the long period of care that *V. spectrum* provide for their offspring) could represent limiting factors in managed environments. The habitat required for the protection of this species should encompass its home range as well as habitat for its diverse prey [61]. Given that prey included mature forest associated species, forest management should be a focus for conservation of this species. Similar patterns for association of roost and foraging habitats in mature forest habitats have been described in other bat species like *M. microtis* and *Pipistrellus subflavus* and other taxa (e.g., birds; [26, 49, 50, 62–65]). Mature and riparian forest might also be needed for other clutter-flying adapted bat species to commute

and forage in a mosaic of agricultural and forested land [25, 66]. In this work, we showed that the combination of DNA metabarcoding and telemetry techniques can improve our understanding of the ecology and community interactions of a rare carnivorous bat species. Novel ecological data on diet, roosts, and habitat use of *V. spectrum* provided additional support for habitat management and conservation efforts of large mature forest patches (> 2000 ha) like Escameca Grande and overall species diversity in the landscape as important components of the spectral bat habitat requirements.

## Supporting information

**S1 File.**
(DOCX)

## Acknowledgments

We thank M. F. Chávez-Velásquez, M. Fernández-Mena, L. E. Gutiérrez-López, C. Hood, B. Noble, M. Parker, and A. Sohikian for field assistance and sample collection and transportation. We also thank Paso Pacífico and personnel C. Bonilla, H. Espinosa-Acevedo, and R. S. Castellón from the Ministerio del Ambiente y Recursos Naturales de Nicaragua (MARENA) for fieldwork logistics and research permits. We thank F. J. Bonaccorso, A. Dikeman, J. Foster, P. Fulé, S. Hershauer, J. Lyman, D. Sanchez, C. Sobek, T. Theimer, J. Upton, and E. Westeen for their training in lab techniques, sample preparation, bioinformatic analysis, and manuscript revision.

## Author Contributions

**Conceptualization:** José Gabriel Martínez-Fonseca, Faith M. Walker, Carol L. Chambers.

**Data curation:** José Gabriel Martínez-Fonseca, Kei Yasuda, Carol L. Chambers.

**Formal analysis:** José Gabriel Martínez-Fonseca, Rebecca Mau, Faith M. Walker.

**Project administration:** Faith M. Walker, Carol L. Chambers.

**Supervision:** José Gabriel Martínez-Fonseca, Faith M. Walker, Kei Yasuda, Carol L. Chambers.

**Writing – original draft:** José Gabriel Martínez-Fonseca, Arnulfo Medina-Fitoria, Carol L. Chambers.

**Writing – review & editing:** José Gabriel Martínez-Fonseca, Faith M. Walker, Arnulfo Medina-Fitoria, Kei Yasuda, Carol L. Chambers.

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
