## [Decision Letter · Decision Letter 0]

19 Jan 2022

PONE-D-21-38967Vampyrum spectrum (Phyllostomidae) space use and prey revealed by radio-telemetry and DNA metabarcodingPLOS ONE

Dear Dr. Martinez-Fonseca,

Thank you for submitting your manuscript to PLOS ONE. After careful consideration, we feel that it has merit but does not fully meet PLOS ONE’s publication criteria as it currently stands. Therefore, we invite you to submit a revised version of the manuscript that addresses the points raised during the review process.

Both reviewers acknowledge that this is an important contribution to understanding the foraging ecology of a rare carnivorous bat. There are a number of helpful clarifying comments and questions that the authors should take into consideration, including more methodological detail. Both reviewers also highlight that the title is inaccurate, as space use (e.g., home range) isn’t something the authors can quantify with the two relocations and one individual. 

We look forward to receiving your revised manuscript.

Kind regards,

Daniel Becker

Academic Editor

PLOS ONE

Journal Requirements:

Reviewers' comments:

Reviewer's Responses to Questions

**Comments to the Author**

1. Is the manuscript technically sound, and do the data support the conclusions?

Reviewer #1: Partly

Reviewer #2: Yes

2. Has the statistical analysis been performed appropriately and rigorously? 

Reviewer #1: N/A

Reviewer #2: N/A

3. Have the authors made all data underlying the findings in their manuscript fully available?

Reviewer #1: Yes

Reviewer #2: No

4. Is the manuscript presented in an intelligible fashion and written in standard English?

Reviewer #1: Yes

Reviewer #2: Yes

5. Review Comments to the Author

Reviewer #1: This research is a great contribution to the knowledge base of a rare and carnivorous bat species, Vampyrum spectrum. The article is well written, and I would like to see a bit more detail in the Methods so that others could replicate the steps taken to analyze fecal samples and to find the roosting tree. For publication in PLOS ONE, this paper could also benefit from some discussion of other carnivorous bat studies, particularly those using eDNA. Adding some additional supporting citations could help reach a broader audience. I also recommend a slight change to the title that better represents the underlying data.

Line 4: Space use may infer home range or foraging range analysis; in this study, there were two relocations (capture site, dropped transmitter) from one individual. Whereas these data points shed some light on two specific foraging locations, they are a very small proportion of the data needed to estimate space use.

Suggest changing ‘space use’ to ‘foraging distance’ or ‘foraging activity’ in the title.

• Also capitalize each word in title

Line 25: add ‘is’ before ‘listed as’

Line 34: The prior line states that primers were used to detect vertebrates and arthropods, but then there is no information about arthropods detected. Were there any obvious large arthropods detected such as katydids or beetles? Even if these weren’t quantified (as stated in the Results section), it would be worth noting here for comparison with other carnivorous bat studies.

Line 39: Change ‘the species’ to ‘Vampyrum spectrum’ for a stronger closing

Line 47: Change ‘associated with’ to ‘inhabit’

Line 49: Change ‘through’ to ‘throughout’

Line 60: Reword this sentence for flow: ‘Feathers found at a communal roost entrance suggest the predation and transport of at least 18 bird species ranging from the 150 g White-tipped Dove (Leptotila verreauxy) to the 20-g Banded Wren (Thryothorus pleurosticus) [2].

• Also, capitalize all bird species common names throughout (line 259)

Line 66: ‘often implies that collection of live specimens is necessary’

Line 70: Add a paragraph about what is known about the diet of neotropical carnivorous bats from metabarcoding studies (e.g., Jones, P. L., Divoll, T. J., Dixon, M. M., Aparicio, D., Cohen, G., Mueller, U. G., ... & Page, R. A. (2020). Sensory ecology of the frog-eating bat, Trachops cirrhosus, from DNA metabarcoding and behavior. Behavioral Ecology, 31(6), 1420-1428.)

Line 71: add ‘to’ before describe; ‘activity patterns’ is a bit vague, consider changing to ‘foraging distance’ or ‘foraging areas’

Line 74: delete ‘for foraging’, add ‘while foraging’ to end of sentence

Line 81: Approximately how many km wide and long is the Rivas Isthmus? This info will help readers visualize the broader landscape

Line 83: Add info about elevation changes or landscape features. I realize it is lowland, but any karst features may make it more challenging to conduct radio telemetry. A bit more detail here could give readers an appreciation for the challenges of tracking bats in that area.

Line 98: With PLOS ONE’s broad readership, it would help to add ‘skin’ or ‘medical’ before the word glue, for those unfamiliar with the common practice of using this type of glue on animals.

Line 99: Is there a citation or some more detail about the ‘close approach method’? More detail will allow others to replicate the method when searching for similar roost types.

Line 100: Was a 3- or 5-element yagi antenna used? This will help others choose the right equipment

Line 107: Describe the start and end time of the exit/entry surveys so readers can guage whether the bats are going out/in several times per night or for one extended foray.

Line 110: Consider an alternate term to ‘≥24 hr’. Is it possible that these samples were defecated less than 24 hr before collection? For example, most of the sample might be days or weeks old, but some might have been deposited 15 min before the samples were collected. Maybe ‘roost sample’ and ‘fresh sample’?

Line 127: Were samples indexed on both ends and were these prepared for paired-end sequencing? How many samples were run simultaneously and were samples from all primer sets combined in the same run?

Line 131: Add some detail about the parameters chosen during bioinformatics. What quality threshold was used for trimming? What copy number threshold was used to determine if OTUs were potential sequencing artifacts? Was there a step to check for chimeras or crosstalk between samples? (Schnell, I. B., Bohmann, K., & Gilbert, M. T. P. (2015). Tag jumps illuminated–reducing sequence-to-sample misidentifications in metabarcoding studies. Molecular Ecology Resources, 15, 1289–1303. https://doi. org/10.1111/1755-0998.12402)

Line 136: Were taxonomic assignments always clear cut to another sister species or backed off to genus in ambiguous cases (i.e., more than one sister species possible in the same genus)?

Line 152: What happened between Mar. 4 and Mar. 14? Was the bat elsewhere or not searched for during that timeframe?

Line 165: ‘due to decomposition’

Line 201: add ‘which range from’ after Gallus gallus to avoid confusion. At first I thought that the results indicated chickens of various sizes in the diet of the bats.

Table 2: Use en dashes between the values in the Mass column

Line 233: Were there any unusually large arthropods detected, such as beetles or katydids? The primer sets should pick them up and it would be worth noting, even if not quantified, for comparison with other studies that have documented V. spectrum eating insects (Bonato et al. 2004).

Line 251: Perhaps note that ANML was the only set to detect the screech owl? Hence, it may be worth using multiple primer sets, when possible, to recover more prey items than one primer set alone.

Line 275: change ‘were’ to ‘are’ when referencing published findings

Line 282: The interesting things about V. spectrum foraging opportunistically on small vertebrates is that they don’t appear to eat anurans or lizards like other carnivorous false-vampires (False Vampires and Other Carnivores - Bat Conservation International). Do you expect that the primer sets would have picked up reptiles if they had eaten them? Some comparison/discussion of the diet of other false vampires (at least neotropical, and possibly African and Australian) would be useful here and appeal to a broader audience.

Line 296: Sheppard et al. 2005 describe this secondary predation phenomenon: Sheppard, S. K., Bell, J., Sunderland, K. D., Fenlon, J., Skervin, D., & Symondson, W. O. C. (2005). Detection of secondary predation by PCR analyses of the gut contents of invertebrate generalist predators. Molecular ecology, 14(14), 4461-4468.

Line 298: ‘longer period ,and, thus’

Line 300: delete duplicate ‘preserve’

Line 314: Add citation for this statement. See Arrizabalaga-Escudero, A., Garin, I., García-Mudarra, J. L., Alberdi, A., Aihartza, J., & Goiti, U. (2015). Trophic requirements beyond foraging habitats: The importance of prey source habitats in bat conservation. Biological Conservation, 191, 512-519.

Line 315: Clarify ‘forest management’ here. Some may assume that forest management = timber harvest, which sounds at odds with mature forests that V. spectrum may require. Are there any management practices that might support large roost trees in mature forest?

References: check italics on species names and en dashes in page number ranges. Also add missing DOIs

Reviewer #2: I assumed by the title that the manuscript would give details about space use by V. spectrum. However, the manuscript describes the time one individual leaves and arrives to the roost, and we have only three locations: capture, roost and one location where the transmitter was dropped. The radiotelemetry was used to find the day roost, in which the feces were collected and the diet was accessed by DNA metabarcoding techniques.

I liked the relation between the prey habitats and the carnivore habitat, and it can generate important insights about the carnivore niche, specially when it is about a difficult to find and monitor species like V. spectrum.

The manuscript is well written and the research about the natural history of the species is well conducted. I tried to help make the text clearer. Please see my specific comments below.

Line 52 – localities where V. spectrum was captured?

Line 56 e 57 – Can you explain more about the relation between foraging activity and resource availability for V. spectrum? As a generalist carnivore, one can assume that this species in not highly impacted by seasonality, for example.

Line 149 and 150 – I do not understand why it was necessary to use two transmitters in a single individual.

Line 157 e 158 – The text is “and six Fresh fecal samples.”. But if it was the first time of accessing this roost, the researchers would not have set the plastic to collect fresh fecal sample yet, right?

I found that information on Line 212. Please, add that information on line 157 as well.

Line 175 – What do you mean by “opportunistically”?

Line 186 – here is “lactating female”, but in the line 142 is post-lactating.

Which is correct?

Line 208 – What defines one fecal sample? If the feces are on the floor and you described above “The guano pile itself was 10 cm deep”.

Table 2: Please, add to the description of the table the meaning of “ANML”, “SF”, “F”, “12S” and “18S”

Line 256 – 259. I do not understand the sentence.

Line 270 – 278. It seems to me that the sentences describing bat prey niches are better placed on line 266, after the sentence “Bat species represented most of the mammal species detected in our samples and represented a range of masses, foraging guilds, and habitat associations.”

Line 300 – delete first “preserve”.

Line 303 – 304. If the female was lactating, the puppy should be with her, right? Why didn`t you see it?

Line 316 – Please explain more about “Similar patterns for association of roost and foraging habitats”. What association?

Line 317 – What species?

Please explain how the activity monitoring was conducted. Does the table 1 describe the first time the individual left the roost during the night? Does the individual leave the roost again after? Do observation nights consisted of being close to the tree recording the time the individual leave and arrive?

The individual was captured on 3 March but the roost was found on 14 March. Would you comment on the difficulty of finding the individual? Was it recorded overnight even without locating the roost?

6. PLOS authors have the option to publish the peer review history of their article (what does this mean?). If published, this will include your full peer review and any attached files.

Reviewer #1: No

Reviewer #2: No

---

## [Author Response · Author response to Decision Letter 0]

31 Jan 2022

Response to Reviewers

Review Comments to the Author

Reviewer #1: 

This research is a great contribution to the knowledge base of a rare and carnivorous bat species, Vampyrum spectrum. The article is well written, and I would like to see a bit more detail in the Methods so that others could replicate the steps taken to analyze fecal samples and to find the roosting tree. For publication in PLOS ONE, this paper could also benefit from some discussion of other carnivorous bat studies, particularly those using eDNA. Adding some additional supporting citations could help reach a broader audience. I also recommend a slight change to the title that better represents the underlying data.

We have made changes throughout the manuscript to address all suggestions. See responses to particular questions bellow. 

Line 4: Space use may infer home range or foraging range analysis; in this study, there were two relocations (capture site, dropped transmitter) from one individual. Whereas these data points shed some light on two specific foraging locations, they are a very small proportion of the data needed to estimate space use.

Suggest changing ‘space use’ to ‘foraging distance’ or ‘foraging activity’ in the title.

• Also capitalize each word in title

Title was modified and words capitalized as suggested. 

Line 25: add ‘is’ before ‘listed as’

Changed as suggested 

Line 34: The prior line states that primers were used to detect vertebrates and arthropods, but then there is no information about arthropods detected. Were there any obvious large arthropods detected such as katydids or beetles? Even if these weren’t quantified (as stated in the Results section), it would be worth noting here for comparison with other carnivorous bat studies.

We have expanded on our reasoning for why we decided to not comment on arthropods/invertebrates. Due to a combination of incomplete reference libraries of invertebrates from the Central American region and Neotropics, confidence in ID to species level was low. As reference libraries become more complete, we believe all raw sequences we are making available in NCBI’s depositories will allow a future re-analysis of the data and provide much more informative results. 

Please see also our responses for comments on line 136 and 233. 

Line 39: Change ‘the species’ to ‘Vampyrum spectrum’ for a stronger closing

Changed as suggested 

Line 47: Change ‘associated with’ to ‘inhabit’

Changed as suggested 

Line 49: Change ‘through’ to ‘throughout’

Changed as suggested 

Line 60: Reword this sentence for flow: ‘Feathers found at a communal roost entrance suggest the predation and transport of at least 18 bird species ranging from the 150 g White-tipped Dove (Leptotila verreauxy) to the 20-g Banded Wren (Thryothorus pleurosticus) [2].• Also, capitalize all bird species common names throughout (line 259)

Sentence reworded as suggested and all bird species’ common names are capitalized now. 

Line 66: ‘often implies that collection of live specimens is necessary’

Changed as suggested 

Line 70: Add a paragraph about what is known about the diet of neotropical carnivorous bats from metabarcoding studies (e.g., Jones, P. L., Divoll, T. J., Dixon, M. M., Aparicio, D., Cohen, G., Mueller, U. G., ... & Page, R. A. (2020). Sensory ecology of the frog-eating bat, Trachops cirrhosus, from DNA metabarcoding and behavior. Behavioral Ecology, 31(6), 1420-1428.)

We added as suggested

Line 71: add ‘to’ before describe; ‘activity patterns’ is a bit vague, consider changing to ‘foraging distance’ or ‘foraging areas’

Changed as suggested

Line 74: delete ‘for foraging’, add ‘while foraging’ to end of sentence

Changed 

Line 81: Approximately how many km wide and long is the Rivas Isthmus? This info will help readers visualize the broader landscape

Information added (18 km)

Line 83: Add info about elevation changes or landscape features. I realize it is lowland, but any karst features may make it more challenging to conduct radio telemetry. A bit more detail here could give readers an appreciation for the challenges of tracking bats in that area.

Information added

Line 98: With PLOS ONE’s broad readership, it would help to add ‘skin’ or ‘medical’ before the word glue, for those unfamiliar with the common practice of using this type of glue on animals.

Word “medical” was added 

Line 99: Is there a citation or some more detail about the ‘close approach method’? More detail will allow others to replicate the method when searching for similar roost types.

Line 100: Was a 3- or 5-element yagi antenna used? This will help others choose the right equipment

Information added (2 elements)

Line 107: Describe the start and end time of the exit/entry surveys so readers can guage whether the bats are going out/in several times per night or for one extended foray.

Information added

Line 110: Consider an alternate term to ‘≥24 hr’. Is it possible that these samples were defecated less than 24 hr before collection? For example, most of the sample might be days or weeks old, but some might have been deposited 15 min before the samples were collected. Maybe ‘roost sample’ and ‘fresh sample’?

We renamed and defined samples collected upon discovery of the roost and after as “Sample A” and “Sample B” 

Line 127: Were samples indexed on both ends and were these prepared for paired-end sequencing? How many samples were run simultaneously and were samples from all primer sets combined in the same run?

We added some details on the DNA analysis section in Methods; however, specific information about methods were already provided in references cited (i.e., reference 37).

Line 131: Add some detail about the parameters chosen during bioinformatics. What quality threshold was used for trimming? What copy number threshold was used to determine if OTUs were potential sequencing artifacts? Was there a step to check for chimeras or crosstalk between samples? (Schnell, I. B., Bohmann, K., & Gilbert, M. T. P. (2015). Tag jumps illuminated–reducing sequence-to-sample misidentifications in metabarcoding studies. Molecular Ecology Resources, 15, 1289–1303. https://doi. org/10.1111/1755-0998.12402)

Additional details about bioinformatic parameters and quality control were added on the DNA analysis section in Methods.

Line 136: Were taxonomic assignments always clear cut to another sister species or backed off to genus in ambiguous cases (i.e., more than one sister species possible in the same genus)?

All items reported at the species level were clear cut matches. On a few occasions the closest match was to a sister species that did not occur in our study area. Taxonomical splits and lack of genetic references from Nicaraguan populations were the main reason of mismatches. Taxonomy was updated in Table 2 to reflect the match of our samples with the GenBank reference sequences. 

Line 152: What happened between Mar. 4 and Mar. 14? Was the bat elsewhere or not searched for during that timeframe?

We monitored the bat from Mar 5 but we were not able to climb the tree and pin point the roost entrance until March 14. A clarification was added. 

Line 165: ‘due to decomposition’

Changed as suggested 

Line 201: add ‘which range from’ after Gallus gallus to avoid confusion. At first I thought that the results indicated chickens of various sizes in the diet of the bats.

Changed as suggested 

Table 2: Use en dashes between the values in the Mass column

Changed as suggested 

Line 233: Were there any unusually large arthropods detected, such as beetles or katydids? The primer sets should pick them up and it would be worth noting, even if not quantified, for comparison with other studies that have documented V. spectrum eating insects (Bonato et al. 2004).

The incompleteness of reference sequence on GenBank for arthropod species in the neotropics was another reason for not expanding on the arthropod diet of V. spectrum. We manually BLASTED all sequences from all markers and no there were no matches for large arthropods at least with the current libraries. We added this comments in this section of the manuscript. 

Line 251: Perhaps note that ANML was the only set to detect the screech owl? Hence, it may be worth using multiple primer sets, when possible, to recover more prey items than one primer set alone.

Comment was added as suggested

Line 275: change ‘were’ to ‘are’ when referencing published findings

Changed as suggested 

Line 282: The interesting things about V. spectrum foraging opportunistically on small vertebrates is that they don’t appear to eat anurans or lizards like other carnivorous false-vampires (False Vampires and Other Carnivores - Bat Conservation International). Do you expect that the primer sets would have picked up reptiles if they had eaten them? Some comparison/discussion of the diet of other false vampires (at least neotropical, and possibly African and Australian) would be useful here and appeal to a broader audience.

Yes, the 12S and 18S marker used should have been capable of picking up reptiles or anurans. We added a paragraph expanding on this idea. 

Line 296: Sheppard et al. 2005 describe this secondary predation phenomenon: Sheppard, S. K., Bell, J., Sunderland, K. D., Fenlon, J., Skervin, D., & Symondson, W. O. C. (2005). Detection of secondary predation by PCR analyses of the gut contents of invertebrate generalist predators. Molecular ecology, 14(14), 4461-4468.

We added this reference and expanded on why we decide not to report arthropods on this study. 

Line 298: ‘longer period ,and, thus’

Added a comma

Line 300: delete duplicate ‘preserve’

Deleted 

Line 314: Add citation for this statement. See Arrizabalaga-Escudero, A., Garin, I., García-Mudarra, J. L., Alberdi, A., Aihartza, J., & Goiti, U. (2015). Trophic requirements beyond foraging habitats: The importance of prey source habitats in bat conservation. Biological Conservation, 191, 512-519.

Citation added 

Line 315: Clarify ‘forest management’ here. Some may assume that forest management = timber harvest, which sounds at odds with mature forests that V. spectrum may require. Are there any management practices that might support large roost trees in mature forest?

Changed to “habitat management and conservation” for clarity 

References: check italics on species names and en dashes in page number ranges. Also add missing DOIs

We double checked the completeness of the reference and added DOIs. We used a reference manager (Mendeley) and “Vancouver” style for formatting the reference as indicated by the author guidelines. Some literature that we cite lack DOIs. 

Reviewer #2: 

I assumed by the title that the manuscript would give details about space use by V. spectrum. However, the manuscript describes the time one individual leaves and arrives to the roost, and we have only three locations: capture, roost and one location where the transmitter was dropped. The radiotelemetry was used to find the day roost, in which the feces were collected and the diet was accessed by DNA metabarcoding techniques.

I liked the relation between the prey habitats and the carnivore habitat, and it can generate important insights about the carnivore niche, especially when it is about a difficult to find and monitor species like V. spectrum.

The manuscript is well written and the research about the natural history of the species is well conducted. I tried to help make the text clearer. Please see my specific comments below.

Line 52 – localities where V. spectrum was captured?

Clarification added, Changed to “capture sites” 

Line 56 e 57 – Can you explain more about the relation between foraging activity and resource availability for V. spectrum? As a generalist carnivore, one can assume that this species in not highly impacted by seasonality, for example.

Although this is an interesting point, we are unclear what the reviewer is suggesting and don’t feel that there is adequate information available on the species to assume it is not impacted by seasonality. 

Line 149 and 150 – I do not understand why it was necessary to use two transmitters in a single individual.

We opted for two transmitters since previously other larger bats have been able to detach their glued transmitters within a few days. The collar was an experiment and we wanted to increase the opportunity to obtain data for as many days as possible. We added a clarification for this decision in the methods section. 

Line 157 e 158 – The text is “and six Fresh fecal samples.”. But if it was the first time of accessing this roost, the researchers would not have set the plastic to collect fresh fecal sample yet, right?

I found that information on Line 212. Please, add that information on line 157 as well.

Clarification on this procedure was added

Line 175 – What do you mean by “opportunistically”?

We meant that it was conducted within the possibilities of our personnel availability and were not conducted on consecutive nights necessarily. We added a clarification on this sentence. 

Line 186 – here is “lactating female”, but in the line 142 is post-lactating.

Which is correct?

We corrected it to Post-lactating. 

Line 208 – What defines one fecal sample? If the feces are on the floor and you described above “The guano pile itself was 10 cm deep”.

In this case we referred as a “fecal sample” a tube with a subsample of ~5ml of fecal matter from the pile. We added this definition to the table legend and methods. 

Table 2: Please, add to the description of the table the meaning of “ANML”, “SF”, “F”, “12S” and “18S”

This was added to the table title

Line 256 – 259. I do not understand the sentence.

We rephrase and restructure the paragraph for clarity. 

Line 270 – 278. It seems to me that the sentences describing bat prey niches are better placed on line 266, after the sentence “Bat species represented most of the mammal species detected in our samples and represented a range of masses, foraging guilds, and habitat associations.”

Sentences were moved as suggested 

Line 300 – delete first “preserve”.

Deleted

Line 303 – 304. If the female was lactating, the puppy should be with her, right? Why didn`t you see it?

We believe that one of the individuals is the “puppy” but at this stage is as big as the parents and might already started feeding on captured prey rather than lactating. The female was post-lactating and female bats don’t carry pups every time they fly. We stated this in the results section. 

Line 316 – Please explain more about “Similar patterns for association of roost and foraging habitats”. What association?

We added a clarification for this

Line 317 – What species?

Examples added

Please explain how the activity monitoring was conducted. Does the table 1 describe the first time the individual left the roost during the night? Does the individual leave the roost again after? Do observation nights consisted of being close to the tree recording the time the individual leave and arrive?

The individual was captured on 3 March but the roost was found on 14 March. Would you comment on the difficulty of finding the individual? Was it recorded overnight even without locating the roost?

We added clarification for all these points in methods and results. The bat monitored left only once per night. We expanded the detail on our methods for activity monitoring in the method section under the ‘live capture and telemetry” sub-heading. We added information on the challenges of tracking the bat in that environment on our results section. 

---

## [Decision Letter · Decision Letter 1]

6 Mar 2022

PONE-D-21-38967R1Vampyrum spectrum (Phyllostomidae) Movement and Prey Revealed by Radio-Telemetry and DNA MetabarcodingPLOS ONE

Dear Dr. Martinez-Fonseca,

Thank you for submitting your manuscript to PLOS ONE. After careful consideration, we feel that it has merit but does not fully meet PLOS ONE’s publication criteria as it currently stands. Therefore, we invite you to submit a revised version of the manuscript that addresses the points raised during the review process.

Congratulations to the authors for a thorough revision. The reviewers only have some minor clarifying revisions, which the authors should be able to readily address. ==============================

We look forward to receiving your revised manuscript.

Kind regards,

Daniel Becker

Academic Editor

PLOS ONE

Journal Requirements:

Reviewers' comments:

Reviewer's Responses to Questions

**Comments to the Author**

1. If the authors have adequately addressed your comments raised in a previous round of review and you feel that this manuscript is now acceptable for publication, you may indicate that here to bypass the “Comments to the Author” section, enter your conflict of interest statement in the “Confidential to Editor” section, and submit your "Accept" recommendation.

Reviewer #1: (No Response)

Reviewer #2: All comments have been addressed

2. Is the manuscript technically sound, and do the data support the conclusions?

Reviewer #1: Yes

Reviewer #2: Yes

3. Has the statistical analysis been performed appropriately and rigorously? 

Reviewer #1: Yes

Reviewer #2: N/A

4. Have the authors made all data underlying the findings in their manuscript fully available?

Reviewer #1: Yes

Reviewer #2: Yes

5. Is the manuscript presented in an intelligible fashion and written in standard English?

Reviewer #1: Yes

Reviewer #2: Yes

6. Review Comments to the Author

Reviewer #1: Comments on the tracked changes version:

1. Line 75 - change "Neotropics DNA techniques" to "Neotropics, DNA metabarcoding"

2. Line 107 - change "the roost" to "roost sites". In the methods, we don't know yet that there is only one roost

3. DNA Collection section - The sample numbers were lost in the updates. Specify how many of Sample A and how many of Sample B

4. Line 147 - How many total samples were in the sequencing run, after PCR (including controls) but before equimolar pooling? This will be important for readers to understand how 16 raw fecal samples equals many more after splitting for each primer set and PCR (e.g., were there duplicate samples, were there PCR failures, was there equal representation from all the primer sets). Some studies may run 96 samples for the convenience of using a 96-well plate during PCR and some multiplex many more into one sequencing run; the results may be different for each sampling strategy.

5. Line 179 - commas before and after "therefore"

6. Line 274 - this idea needs a bit more clarification. Using the potential to detect prey that arthropods had eaten is not not a strong justification as there may have been secondary detections from birds or bats eating arthropods or secondary bird or mammal detections in vampire bats. I think for this paper it is fine to not report arthropod detections in detail, the justification just needs a bit of clarification. Can you present a proportion of ASVs? If arthropods were a small proportion of ASVs, then that may be justification for not reporting those results. The BOLD database and GenBank have a lot of arthropod sequences for the COI barcode, and though the exact Neotropical species may not be represented, results would come back in the same genera or families. It would be useful to know if there were any arthropods that were detected in most samples or if it was a selection of low frequency results from various arthropod orders.

Reviewer #2: This research is a great contribution to the knowledge base of a rare and carnivorous bat species, Vampyrum spectrum. All the comments have been addressed and the manuscript is clear and well-written.

7. PLOS authors have the option to publish the peer review history of their article (what does this mean?). If published, this will include your full peer review and any attached files.

Reviewer #1: No

Reviewer #2: No

---

## [Author Response · Author response to Decision Letter 1]

10 Mar 2022

Dear Editor and Reviewers: 

Here we resubmit a revised version of the manuscript titled: “Vampyrum spectrum (Phyllostomidae) movement and prey revealed by radio-telemetry and DNA metabarcoding”. All comments and suggestions from reviewers were addressed and accepted as indicated in detail in the sections after this letter. 

Additionally, we have previously uploaded PLOS’ questionnaire on inclusivity in global research in your revised manuscript as supplemental material with this submission. 

Responses to Reviewers (second round or revisions)

Reviewer #1: Comments on the tracked changes version:

1. Line 75 - change "Neotropics DNA techniques" to "Neotropics, DNA metabarcoding"

Changed as suggested

2. Line 107 - change "the roost" to "roost sites". In the methods, we don't know yet that there is only one roost

Changed as suggested

3. DNA Collection section - The sample numbers were lost in the updates. Specify how many of Sample A and how many of Sample B

Changed as suggested, subsample numbers were re included in the text. 

4. Line 147 - How many total samples were in the sequencing run, after PCR (including controls) but before equimolar pooling? This will be important for readers to understand how 16 raw fecal samples equals many more after splitting for each primer set and PCR (e.g., were there duplicate samples, were there PCR failures, was there equal representation from all the primer sets). Some studies may run 96 samples for the convenience of using a 96-well plate during PCR and some multiplex many more into one sequencing run; the results may be different for each sampling strategy.

We added a clarification on this. We used 4 sets of replicates of the original 16 samples to be use with each primer set (4 primers) for a total of 64 replicated samples. Each primer set required different conditions in PCR runs, so we only ran PCRs one replicate-primer at a time on their own plate with their own NTCs and PTCs. If we count the samples and controls, there were a total of 72 data points in 4 plates. We did not experience failures in the PCR runs. 

5. Line 179 - commas before and after "therefore"

Changed as suggested

6. Line 274 - this idea needs a bit more clarification. Using the potential to detect prey that arthropods had eaten is not not a strong justification as there may have been secondary detections from birds or bats eating arthropods or secondary bird or mammal detections in vampire bats. I think for this paper it is fine to not report arthropod detections in detail, the justification just needs a bit of clarification. Can you present a proportion of ASVs? If arthropods were a small proportion of ASVs, then that may be justification for not reporting those results. The BOLD database and GenBank have a lot of arthropod sequences for the COI barcode, and though the exact Neotropical species may not be represented, results would come back in the same genera or families. It would be useful to know if there were any arthropods that were detected in most samples or if it was a selection of low frequency results from various arthropod orders.

We think this is an interesting question but beyond the scope of this paper. We found 13% of ASVs represented arthropods (22 Orders) in our sample. They represented leafhoppers, flies, cockroaches, micro-caddisflies, grass flies, copepods, and moths. However, we did not feel confident concluding they were potential arthropod diet items from V. spectrum. We did address the fact that prey diet could be picked up from the host’s fecal material and had already addressed this in the document. We believe that the data on small vertebrates presented in the manuscript summarizes the most likely diet items of our study carnivore species. 

 We believe that a comprehensive analysis of arthropods should involve sampling and barcoding of these on our study site. In our manuscript we briefly mentioned arthropod orders found in the samples and published all raw ASV data from all primer sets for future research in this topic (line 261-271; 318-329).

Reviewer #2: This research is a great contribution to the knowledge base of a rare and carnivorous bat species, Vampyrum spectrum. All the comments have been addressed and the manuscript is clear and well-written.

---

## [Editor Report · Decision Letter 2]

11 Mar 2022

Vampyrum spectrum (Phyllostomidae) Movement and Prey Revealed by Radio-Telemetry and DNA Metabarcoding

PONE-D-21-38967R2

Dear Dr. Martinez-Fonseca,

We’re pleased to inform you that your manuscript has been judged scientifically suitable for publication and will be formally accepted for publication once it meets all outstanding technical requirements.

Kind regards,

Daniel Becker

Academic Editor

PLOS ONE

Additional Editor Comments (optional):

I thank the authors for addressing all the reviewer concerns, and I look forward to seeing this paper online. It's an important contribution to the ecology of a rare carnivorous bat.
---

## [Editor Report · Acceptance letter]

17 Mar 2022

PONE-D-21-38967R2 

*Vampyrum spectrum* (Phyllostomidae) Movement and Prey Revealed by Radio-Telemetry and DNA Metabarcoding. 

Dear Dr. Martinez-Fonseca:

I'm pleased to inform you that your manuscript has been deemed suitable for publication in PLOS ONE. Congratulations! Your manuscript is now with our production department. 

Kind regards, 

on behalf of

Dr. Daniel Becker 

Academic Editor

PLOS ONE